# Visible Near-Infrared Photodetection Based on Ta_2_NiSe_5_/WSe_2_ van der Waals Heterostructures

**DOI:** 10.3390/s23094385

**Published:** 2023-04-29

**Authors:** Pan Xiao, Shi Zhang, Libo Zhang, Jialiang Yang, Chaofan Shi, Li Han, Weiwei Tang, Bairen Zhu

**Affiliations:** 1College of Science, Zhejiang University of Technology, 288 Liu-He Road, Hangzhou 310023, China; 2College of Physics and Optoelectronic Engineering, Hangzhou Institute for Advanced Study, University of Chinese Academy of Sciences, No. 1, Sub-Lane Xiangshan, Xihu District, Hangzhou 310024, China; 3State Key Laboratory for Infrared, Physics Shanghai Institute of Technical Physics, Chinese Academy of Sciences, 500 Yu-Tian Road, Shanghai 200083, China

**Keywords:** Ta_2_NiSe_5_, WSe_2_, heterostructure, Schottky barrier, photodetection, imaging

## Abstract

The increasing interest in two-dimensional materials with unique crystal structures and novel band characteristics has provided numerous new strategies and paradigms in the field of photodetection. However, as the demand for wide-spectrum detection increases, the size of integrated systems and the limitations of mission modules pose significant challenges to existing devices. In this paper, we present a van der Waals heterostructure photodetector based on Ta_2_NiSe_5_/WSe_2_, leveraging the inherent characteristics of heterostructures. Our results demonstrate that this detector exhibits excellent broad-spectrum detection ability from the visible to the infrared bands at room temperature, achieving an extremely high on/off ratio, without the need for an external bias voltage. Furthermore, compared to a pure material detector, it exhibits a fast response and low dark currents (~3.6 pA), with rise and fall times of 278 μs and 283 μs for the response rate, respectively. Our findings provide a promising method for wide-spectrum detection and enrich the diversity of room-temperature photoelectric detection.

## 1. Introduction

The photoelectric industry has been experiencing continuous development over the past few decades and photoelectric devices have become ubiquitous in our daily lives. As the core of photoelectric devices, the photo-detector is a type of application equipment that converts optical signals into electrical signals. It is a key component of equipment that is frequently used in various fields such as medical diagnosis, optical communication, security monitoring, and military applications [1,2,3,4,5]. Conventional photodetectors, including Si, SiC, GaN, and ZnO [6,7], exhibit excellent single-band optical responses due to their band gaps. Silicon-based materials are generally used in visible light detectors, which have a wide range of applications in color imaging for realistic everyday scenes, while wide-bandgap semiconductor materials such as SiC, GaN, and ZnO are more suitable for UV detectors and have a wide range of application scenarios for anti-missile and tracking in the solar blind range of ultraviolet light [8,9]. However, as the application of photodetectors expands, the need to identify complex working environmental information is increasing. In the face of wide-spectrum detection, multiple detectors of different bands are often required to perform together; however, there has been a gradual failure to meet the current needs of the actual photoelectric application field. Therefore, the research on and development of broadband photodetectors that can cover multi-band response is particularly urgent at present [10,11,12].

Over the past decade, two-dimensional (2D) materials have gradually attracted considerable attention due to their atomic thin thickness, bandgap tunability, and easy integration of van der Waals (vdWs) heterostructures [13,14,15,16,17]. In particular, graphene possesses a zero bandgap and high carrier mobility, making it an attractive material for broadband response research. However, its natural spectrum absorption is only 2.3% and its zero bandgap results in relatively low response performance and limitations in photodetection research [18]. While single-layer MoS_2_ has demonstrated superior performance in the visible spectrum range, its application in the infrared spectrum of equal width is significantly restricted by its large bandgap [19,20,21,22]. Although black phosphorus (BP) exhibits a tunable narrow-bandgap (0.3–1.5 eV) and high carrier mobility, its poor stability currently impedes its widespread utilization in photoelectric detection research [23,24,25,26]. Recently, terephthalic thiocompound (Ta_2_NiSe_5_), a layered material with weak van der Waals force interactions, has shown significant potential in wide-band photodetection [27,28]. Unlike other two-dimensional materials, Ta_2_NiSe_5_ is a ternary chalcogenide compound that possesses a direct narrow bandgap (E_g_ ~ 0.33 eV) [29,30,31,32]. However, the Ta_2_NiSe_5_ photodetector suffers from the drawbacks of excessive dark current and high power consumption, which restrict its wide application in the field of photodetection [33]. The fabrication of heterogeneous devices may offer a potential solution to these problems associated with Ta_2_NiSe_5_. Heterojunctions based on two-dimensional materials can achieve self-powered, high photo response and a good signal ratio, which is determined by the mechanism of photocurrent generation. Moreover, in some structures, the appearance of heterogeneous interface Schottky barriers generates a built-in electric field that separates the electron-hole pairs generated by light, thereby inhibiting the rapid recombination of photocarriers [34,35,36].

In this work, we fabricated a photodetector by constructing a Ta_2_NiSe_5_/WSe_2_ van der Waals heterostructure on a metal electrode. A series of experimental data showed that Ta_2_NiSe_5_/WSe_2_ heterojunctions not only reduce the dark current and power consumption of single-material photodetectors by utilizing the Schottky barrier formed but also address the issue of limited detection bands in single-material photodetectors, thereby enabling wide-spectral detection from visible to infrared light. They also show good application potential in the field of visible light imaging. Our experimental results confirm the feasibility of Ta_2_NiSe_5_/WSe_2_ heterojunction photodetectors for wide-spectrum detection, low power consumption, and fast response.

## 2. Device Design and Fabrication

We obtained bulk Ta_2_NiSe_5_ and WSe_2_ single crystals from Shanghai Onway Technology Co., Ltd. (Shanghai, China). Few-layer Ta_2_NiSe_5_ and WSe_2_ were exfoliated using scotch tape and then assembled using the all-dry transfer method to form van der Waals heterojunctions. As illustrated in Figure 1, the Ta_2_NiSe_5_/WSe_2_ heterostructure was constructed using the all-dry transfer method, which shows the process of heterojunction fabrication. Initially, we mechanically peeled off Ta_2_NiSe_5_ sheets of appropriate size using tape, followed by placing polydimethylsiloxane (PDMS) as an intermediate carrier onto the tape with the materials during the transfer process. Subsequently, we oriented the other side of the PDMS towards the substrate, with the material side facing down, to transfer the Ta_2_NiSe_5_ sheet onto the substrate. To transfer the WSe_2_ sheets, a similar technique was employed, but with the added requirement of operating and observing the specific position under a microscope in real-time to ensure the successful stacking of the WSe_2_ sheet onto the Ta_2_NiSe_5_ sheet to obtain a complete Ta_2_NiSe_5_/WSe_2_ heterojunction.

In this study, we investigated the photoelectric performance of the Ta_2_NiSe_5_/WSe_2_ heterojunction device under various environmental conditions in real-time, as illustrated in Figure 2a. Initially, Cr/Au (10 nm/50 nm) metal electrodes were deposited onto a low-resistance silicon substrate coated with SiO_2_ (300 nm) using UV lithography technology. Next, the Ta_2_NiSe_5_ sheet, prepared by mechanical exfoliation, was placed on the substrate via the standard PDMS pressing method, with the same preparation transfer process for the WSe_2_ sheet, to create a Ta_2_NiSe_5_/WSe_2_ heterostructure. The experimental materials were characterized using optical microscopy (Olympus BX51M), scanning electron microscopy with X-ray energy dispersive spectroscopy (SEM-EDS) (Zeiss/GEMINISEM 360), and microscopic laser confocal Raman spectroscopy (Dugout/LABRAM ODYSSEY). Figure 2b presents the optical image of the Ta_2_NiSe_5_/WSe_2_ heterostructure-based photodetector, with A–D denoting the four electrode directions. Different electrodes were connected to apply voltage for the performance testing of various devices. The scanning electron microscopy (SEM) image and corresponding energy dispersive spectra (EDS) mappings are shown in Figure 2c. The Raman spectra of the Ta_2_NiSe_5_, WSe_2_, and Ta_2_NiSe_5_/WSe_2_ heterojunction are represented in Figure 2d. For pure Ta_2_NiSe_5_ [37], three characteristic peaks located at 97.7, 121.8, and 288.8 cm^−1^, attributed to the A_g_ vibration mode, can be clearly observed and are denoted by the orange line. For pure WSe_2_ [38], the Raman spectrum exhibited two distinct diffraction peaks at 247.3 and 257.6 cm^−1^, corresponding to the E_2g_ and A_1g_ lattice vibrational modes in WSe_2_, respectively (red line). It has been reported that monolayers of WSe_2_ do not produce the characteristic peak at 308.3 cm^−1^ under 532 nm laser excitation [39], and the presence of this peak suggests that the exfoliated WSe_2_ flakes were less laminated. All the vibration modes of the Ta_2_NiSe_5_/WSe_2_ heterojunction, regardless of whether Ta_2_NiSe_5_ or WSe_2_ was used (blue line), were clearly observed in the overlapping region, indicating the effective preparation of the Ta_2_NiSe_5_/WSe_2_ heterojunction. According to previous reports, the conductance band (E_c_) and valence band (E_v_) of Ta_2_NiSe_5_ (WSe_2_) are located at −4.6 eV (−4.2 eV) and −5.28 eV (−4.93 eV) [40,41,42,43,44], respectively, forming the type I band alignment of the heterostructure. Figure 2e shows the band structure before and after contact. When the two materials came into contact to form the heterojunction, electrons flowed from Ta_2_NiSe_5_ to the latter due to a higher Fermi energy level than that for Ta_2_NiSe_5_, resulting in a depletion layer near Ta_2_NiSe_5_ and an accumulation layer on the other side, gradually forming an internal electric field [45,46]. The direction of the electric field went from Ta_2_NiSe_5_ to WSe_2_. Applying a forward bias reduced the barrier, facilitating the flow of electrons from Ta_2_NiSe_5_ to WSe_2_, resulting in an increase in current. Conversely, under reverse bias, the barrier increased, impeding the movement of electrons and causing a decrease in current, as illustrated in Figure 2f.

## 3. Results and Discussion

Before discussing the photoelectric performance of the Ta_2_NiSe_5_/WSe_2_ heterojunction detectors, certain photoelectric properties of the individual detector unit are examined to emphasize the advantages of heterojunction detectors. The external circuit was connected to electrodes B and D to conduct the photoelectric performance test of the few-layered WSe_2_ detector. Figure 3a displays the current–voltage (*I*–*V*) curve of the few-layered WSe_2_ photodetector in the dark, which exhibited Schottky characteristics at room temperature and provided a built-in electric field in the device (Mstarter 200). The optoelectronic properties of the few-layered WSe_2_ photodetector were evaluated by varying the incident power (*P*) of 638 nm visible radiation, as shown in Figure 3b. Additionally, Figure 3c depicts the time-resolved photo responses of the various incident powers collected by high-speed oscilloscopes at room temperature without any bias voltage, where the photocurrent *I*_ph_ and the current density *J* are obtained using Equations (1) and (2):(1)$$I_{\mathrm{ph}}=I_{\mathrm{ds}}-I_{\mathrm{dark}}$$
(2)$$J=I_{\mathrm{ds}}/S$$
where *I*_ds_ represents the source-drain current under illumination, *I*_dark_ represents the source-drain current in the dark, and *S* represents the effective area of the device. To elucidate the photocurrent generation mechanism of the few-layered WSe_2_ photodetector, scanning photocurrent microscopy mapping was performed without bias voltage under 638 nm laser irradiation at room temperature. As observed in Figure 3d, the area of the device was mainly distributed between the material and the metal electrode, and there was clear carrier movement within the device, displaying excellent photo response performance. The intrinsic Schottky junction facilitated the rapid separation and collection of electron-hole pairs, producing a photovoltaic current at visible radiation. However, due to the large bandgap limitation of WSe_2_, the detector failed to detect the photo response in the infrared band.

To unveil the photo response of the few-layered Ta_2_NiSe_5_ device in the infrared band, we characterized its optical image by switching on electrodes A and C, as shown in the inset of Figure 4a. In contrast to the WSe_2_ device, the current–voltage (*I*–*V*) curve of the Ta_2_NiSe_5_ device exhibited excellent linearity in the dark, as depicted in Figure 4a. Here, the dark current was measured to be 126 uA at a bias voltage of −0.1 V. Figure 4b presents the linear current–voltage output curve of the Ta_2_NiSe_5_ device under illumination of 1550 nm. However, due to a large dark current, the photocurrent was not very apparent, even after applying the bias voltage. To further assess the photo response at 1550 nm, we measured the time-resolution photo response under periodic illumination with a fixed bias (*V*_ds_ = 0.1 V) by switching to different optical powers (see Figure 4c, which shows a significant power dependency). In the near-infrared band, since the energy of the incident photon exceeded the bandgap of Ta_2_NiSe_5_ (hv > E_g_), the photoinduced electron-hole pairs were separated by an external electric field and collected by the electrodes. The scanning photocurrent microscopy mapping revealed that the photocurrent region was far from the contact region of the electrode and Ta_2_NiSe_5_ material. It is noteworthy that, although the Ta_2_NiSe_5_ material is capable of realizing near-infrared photodetection, it is uncompetitive for future applications due to the large dark current caused by bias in photoconductivity mode.

The experimental results of the detectors based on the two individual materials mentioned above led to the proposal of a novel Ta_2_NiSe_5_/WSe_2_ vdWs heterogeneous structure that achieves low dark current levels from visible to infrared detection. During the testing process, we varied the wiring configuration by connecting the source and drain electrodes at terminals C and D. The current–voltage (*I*–*V*) output curve of the vdWs heterogeneous device at room temperature with and without laser illuminations of 638 nm and 1550 nm, respectively, is presented in Figure 5a. By increasing the light incident power of 638 nm and 1550 nm, the Ta_2_NiSe_5_/WSe_2_ heterojunction device exhibited excellent photo response performance, as shown in Figure 5b,c, where a heterojunction is clearly visible. The phenomenon of negative short-circuit current (~4.3 pA) and positive open-circuit voltage (~0.1 V), as revealed in Figure 5a–c, is indicative of a photovoltaic-type mode. At a bias voltage of −1 V, the dark current of the proposed device was measured to be 3.6 pA, which is eight orders of magnitude lower than that of the Ta_2_NiSe_5_ device. The presence of the heterojunction not only reduced the dark current and improved the signal-to-noise ratio but also enhanced the light on/off ratio of the photocurrent. Figure 5d,e demonstrate the time-resolved photo response without any external electric field and with increasing incident power at 638 nm and 1550 nm, respectively, showing a fast photo-switching process. To further confirm the photo response mechanism, the location of the scanning photocurrent mapping focused on the junction area (see Figure 5f), which confirmed the origin of the photocurrent at the heterojunction. In the heterojunction region, electrons in Ta_2_NiSe_5_ were excited to the conduction band by IR radiation, generating electron-hole pairs, which were separated by the built-in electric field and flowed towards WSe_2_, being ultimately collected by electrodes. Conversely, holes flowed in the opposite direction, forming a circuit and producing photocurrent.

As illustrated in Figure 6a,b, photocurrent was observed in both the visible and infrared bands of the Ta_2_NiSe_5_/WSe_2_ heterojunction device, without applying any bias. Laser irradiation of the Ta_2_NiSe_5_/WSe_2_ heterojunction device generated photogenerated electrons, which then moved directionally to form photocurrent under the influence of the built-in electric field within the device. To evaluate the crucial parameters of the Ta_2_NiSe_5_/WSe_2_ heterojunction photodetector, responsivity *R*, characterized as the value of the photocurrent generated per unit of optical power, is defined as follows [47]:(3)$$R=I_{\mathrm{ph}}/P_{\mathrm{in}}$$

Specific detectivity *D*^*^ is another important parameter of the photodetector and represents the sensitivity of the photodetector; it is defined as follows [48]:(4)$$D^{*}=R/{\left(2qI_{dark}/S\right)}^{0.5}$$
where *I*_ph_ is the photocurrent, Pin is incident light power, *q* is the elementary charge, and *S* is the illumination active area on the device. For the visible/IR band, the extracted power-dependent responsivity and detectivity are plotted in Figure 6c,d. The responsivity of the heterojunction device was ~0.04 mA/W (*P* = 11.12 μW) at 638 nm and ~0.82 nA/W (*P* = 231.8 μW) at 1550 nm. The detectivity of the heterojunction device was approximately 3.6 × 10^5^ Jones (*P* = 11.12 μW) at 638 nm and approximately 90 Jones (*P* = 231.8 μW) at 1550 nm. Figure 6e presents the time-resolved optical response at 638 nm in the absence of any external electric field, revealing a rapid optical switching process. During this period, the photocurrent response of the device remained relatively stable, indicating good stability of the heterojunction device.

Response time, another key parameter, which is routinely defined as the time required by the photo response to rise from 10% to 90% or fall from 90% to 10% in a single impulse, is retrieved directly from the high-speed sampling oscilloscope (Tektronix/Afg 3100 series) [49]. The rise and fall times of the heterojunction device were 278 μs and 283 μs, respectively, as shown in Figure 6f. We also compared the performance test results with other van der Waals heterojunction photodetectors, as shown in Table 1 [50,51,52,53,54]. The comparison revealed that the Ta_2_NiSe_5_/WSe_2_ heterojunction detectors prepared in our experiment exhibit superior performance in terms of dark current and response times, surpassing previously reported photodetectors.

In addition, in order to explore the potential application of the Ta_2_NiSe_5_/WSe_2_ heterojunction photodetectors in the imaging field, we carried out imaging experiments on the detectors. Imaging was conducted using the visible spectrum (638 nm) on the Ta_2_NiSe_5_/WSe_2_ heterojunction photoelectric device, as illustrated in Figure 7a (equipment type: THORLABS CM401). Figure 7b depicts a schematic diagram of the overall device used in the imaging system. Prior to testing, two pieces of white paper with a certain thickness were prepared, upon which letter graphics (ZJUT, HIAS) were cut out for presentation. During the test, visible light emitted directly at the paper was reflected, while visible light shining on the letter spaces penetrated.

## 4. Conclusions

In conclusion, we prepared a van der Waals heterostructure device based on Ta_2_NiSe_5_/WSe_2_ material by stacking thin sheets of Ta_2_NiSe_5_ and WSe_2_ on a SiO_2_ substrate. The resulting heterojunction photodetector exhibited wide-spectrum detection from visible to infrared light, with a response rate of 0.04 mA/W at 638 nm (0.82 nA/W at 1550 nm), utilizing the photoelectric response characteristics of the Ta_2_NiSe_5_/WSe_2_ heterojunction. Furthermore, the device demonstrated extremely low dark current (~3.6 pA) due to the built-in electric field within the van der Waals heterojunction structure, effectively inhibiting dark current generation and promoting the separation of photogenerated electron-hole pairs, resulting in reduced power loss. In addition, the rapid transport and separation of photogenerated carriers under an electric field enabled the Ta_2_NiSe_5_/WSe_2_ heterojunction device to respond quickly, with an up time of approximately 278 μs and a down time of approximately 283 μs. Our results suggest that the van der Waals heterojunction photodetector based on Ta_2_NiSe_5_/WSe_2_ has immense potential for achieving high detection performance, such as broadening the detection band and improving response times.

## Figures and Tables

**Figure 1 sensors-23-04385-f001:**
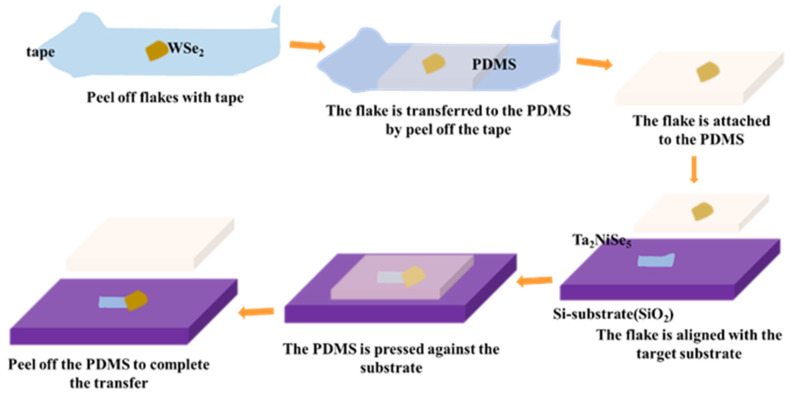
Schematic diagram of heterogeneous structure building process based on Ta_2_NiSe_5_/WSe_2_ material.

**Figure 2 sensors-23-04385-f002:**
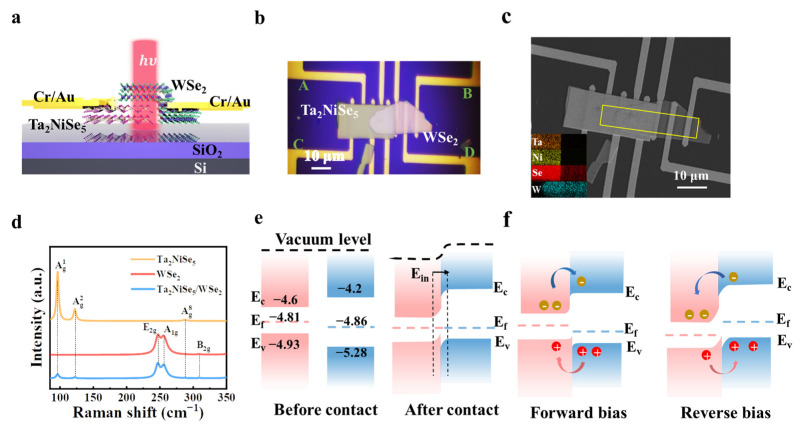
(**a**) Device structure of the photodetector based on the Ta_2_NiSe_5_/WSe_2_ heterostructure. (**b**) The optical image of the photodetector based on the Ta_2_NiSe_5_/WSe_2_ heterostructure. (**c**) SEM image of the Ta_2_NiSe_5_/WSe_2_ heterostructure on the Si/SiO_2_ substrate, with a scale bar of 10 μm. Corresponding EDS mappings of the device are marked by a yellow square. (**d**) Raman spectra of the Ta_2_NiSe_5_, WSe_2_, and Ta_2_NiSe_5_/WSe_2_ overlapped regions. (**e**) Energy band diagram of Ta_2_NiSe_5_/WSe_2_ before and after heterostructure formation. (**f**) Schematic of the band diagram of the heterostructure under forward and reverse bias conditions.

**Figure 3 sensors-23-04385-f003:**
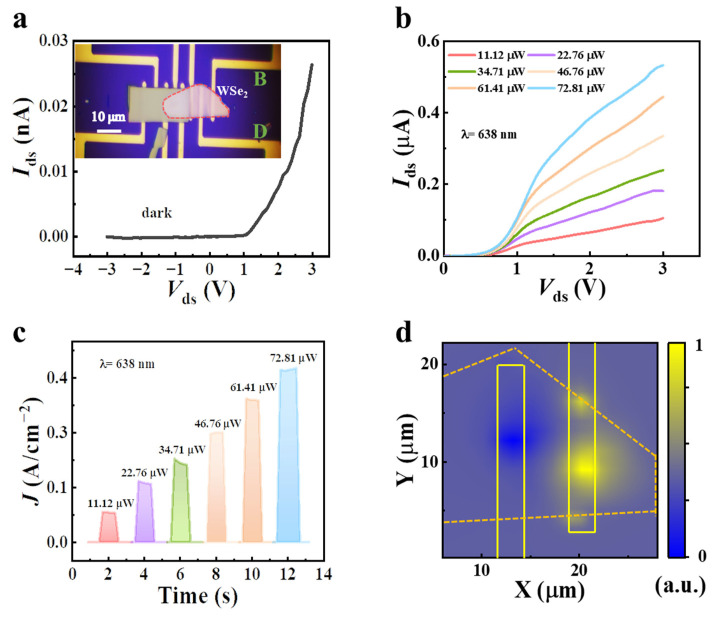
(**a**) Current–voltage characteristics of the WSe_2_ device (inset shows an optical microscopy image of the WSe_2_ device). (**b**) Current–voltage characteristics of the device under the 638 nm laser illumination with different incident powers. (**c**) Time-resolved photo response under different incident powers under 638 nm laser illumination (J denotes the current density). (**d**) Photocurrent mapping of the WSe_2_ device under biases of 0 V with an incident power of 72.81 μW at 638 nm.

**Figure 4 sensors-23-04385-f004:**
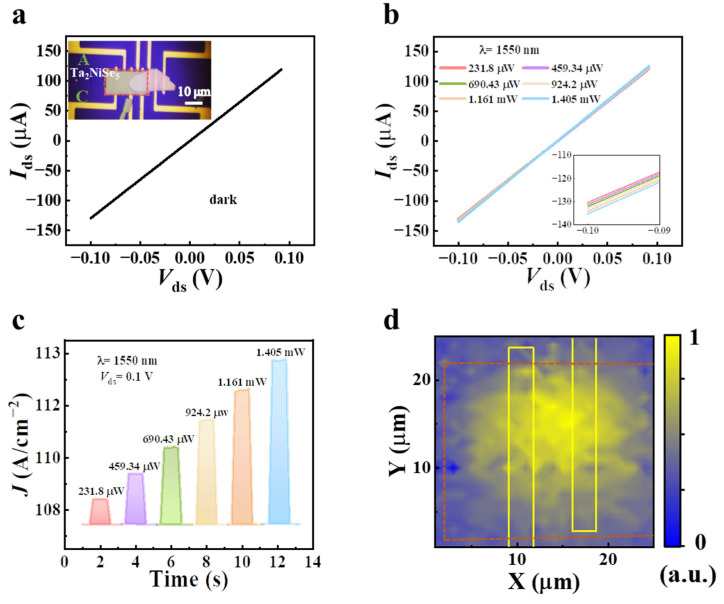
(**a**) Current–voltage characteristics of the Ta_2_NiSe_5_ device (inset shows an optical microscopy image of the Ta_2_NiSe_5_ device). (**b**) Current–voltage characteristics of the device under the 1550 nm laser illumination with different incident powers (inset shows local diagram). (**c**) Time-resolved photo response under different incident powers under 1550 nm laser illumination (*V*_ds_ = 0.1 V). (**d**) Photocurrent mapping of Ta_2_NiSe_5_ photodetector under biases of 0.1 V with an incident power of 1.405 mW at 1550 nm.

**Figure 5 sensors-23-04385-f005:**
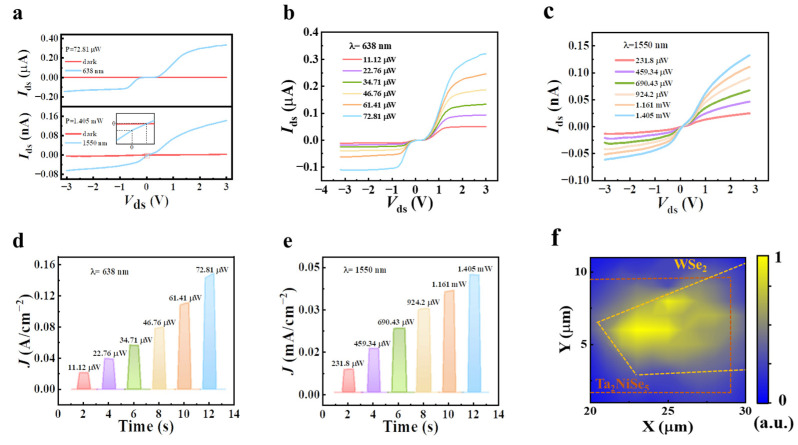
(**a**) Current–voltage characteristics curve in the dark and under 638 and 1550 nm wavelength laser illumination (inset shows local diagram). (**b**,**c**) Current–voltage characteristics of the device under different incident powers under 638 (**b**) and 1550 (**c**) nm laser illumination. (**d**,**e**) Time-resolved photo response under different incident powers under 638 (**d**) and 1550 (**e**) nm laser illumination. (**f**) Photocurrent mapping of Ta_2_NiSe_5_/WSe_2_ photodetector under biases of 0 V with an incident power of 1.405 mW at 1550 nm.

**Figure 6 sensors-23-04385-f006:**
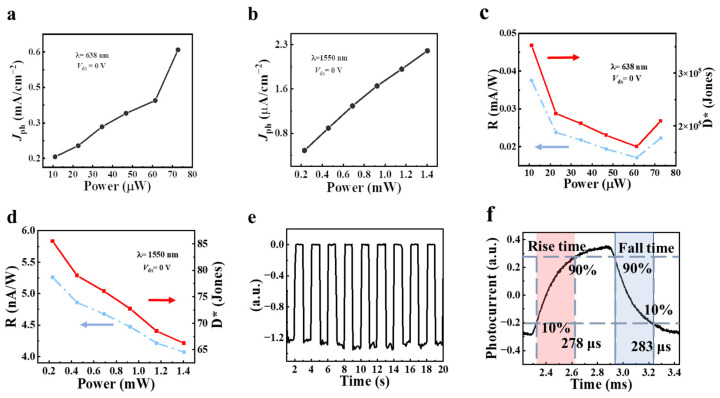
(**a**,**b**) The photocurrent density dependence of the incident power of the Ta2NiSe5/WSe2 photodetector under 638 (**a**) and 1550 (**b**) nm laser illumination (*V*_ds_ = 0 V). (**c**,**d**) Dependence of responsivity and detectivity on incident power under 638 (**c**) and 1550 (**d**) nm laser illumination (*V*_ds_ = 0 V). (**e**) Time-resolved photo response of the heterostructure under 638 nm laser illumination (*P* = 72.81 μW, *V*_ds_ = 0 V). (**f**) Characteristic response times at rise (left) and fall (right) edges.

**Figure 7 sensors-23-04385-f007:**
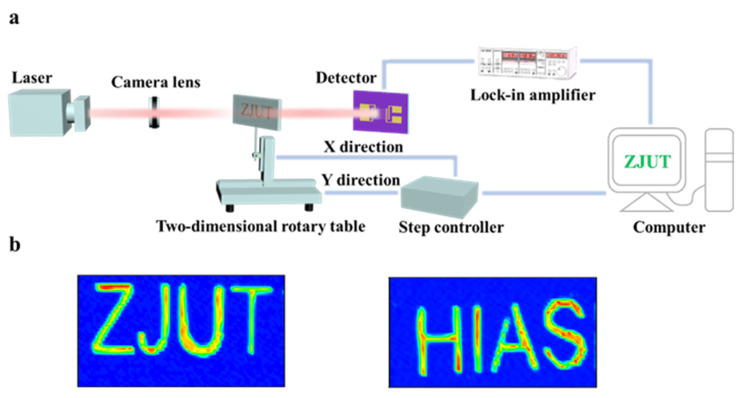
(**a**) Diagram of imaging device. (**b**) The resulting images of “ZJUT” and “HIAS” under 638 nm illumination, respectively.

**Table 1 sensors-23-04385-t001:** Comparison of the performance of photodetectors based on heterojunctions examined in this work and in previous work.

Materials	Wavelength Range (nm)	Dark Current	Rise/Decay Time	On/Off	Reference
Ta_2_NiSe_5_/WSe_2_	638/1550	~3.6 pA	278/283 μs	~10^4^	This work
Ta_2_NiSe_5_/GaSe	520/1550	~4 pA	340/32 ms	5 × 10^3^	[50]
Ta_2_NiSe_5_/MoS_2_	532/1064	~11 pA	7.4/31.1 s	1.9 × 10^2^	[51]
WSe_2_/Si	670	-	75/125 ms	3.2	[52]
WSe_2_/MoSe_2_	532	~0.078 nA	4.3/22.6 ms	~10^3^	[53]
PdSe_2_/Si	532/1550	~10 pA	49/90 ms	~10^3^	[54]

## Data Availability

Not applicable.

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
