# Peer review of "Visible Near-Infrared Photodetection Based on Ta2NiSe5/WSe2 van der Waals Heterostructures"

_sensors, 2023, doi:10.3390/s23094385_

Round 1

Reviewer 1 Report

The present manuscript reports the Vis-NIR photodetectors based on TaNiSe5/WSe2 van der Waals heterostructures. Even though the work is novel, it needs major revision in terms of the device fabrication process parameters and analysis of the results. The following points must be addressed before the manuscript can be accepted for publication.

1. The authors have claimed that the mechanically exfoliated sheets are 2-dimensional in nature without giving any substantial evidence for the same. The electron microscopy images of the 2-D sheets should be included.

2. The phase purity of the sheets should be checked with a diffraction technique and Raman spectroscopy.

3. More details about the device fabrication including all the process parameters must be included.

4. The band offset together with an estimate of the built-in electric field should be included.

5. The comparison of the photodetection characteristics (dark current, response time, etc.) with the existing literature for other van der Waal based heterojunction should be included.

Reviewer 2 Report

Reviewer’s comments:

1.    The cited references should be enhanced for more detail discussion to confirm the research objective of this paper.

2.    In the last paragraph of the “Introduction” section, the research objective and research motivation should be retained, and the experimental data of this research should be deleted due that it has been revealed in “Abstract” and “Results and Discussion” sections.

3.    The research data in this paper should be compared with many other cited literatures using a Table.

4.    The strain effect of heterostructure should be considered and explored in manuscript using a clearly band gap diagram with the carrier transition mechanism.

5.    The well-defined research data including low dark current and high responsivity, as revealed pA and ms respectively, should be clearly described in the content for readability.

6.    In the “Conclusion” section, it is suggested that the research data could be deleted which has been exhibited in the “Abstract” section.

7.    Also, the accuracy, linearity, sensitivity and stability of photoreceiver should be discussed in the manuscript for evaluating the standard performance on the WSe2 device, Ta2NiSe5 device and the Ta2NiSe5/WSe2 heterostructure.

8.    The references from 2022 to 2023 should be cited for the integration of literatures references.

Reviewer 3 Report

The paper presents the results of a study of the photovoltaic properties of the heterostructure created between the layers of Ta2NiSe5 and WSe2 due to Van der Waals forces. The manuscript is presented with a large number of inaccuracies in the design of drawings and descriptions of techniques.

Basic questions:

1. What is the reproducibility of photovoltaic characteristics from sample to sample?

2. The parameters of the photocurrent must be shown in terms of current density per unit contact area. This should be counted in the manuscript.

3. The description of measurement methods in the manuscript is not enough, so it is extremely difficult to understand the purpose of the research. The manuscript does not specify the manufacturers of materials and measuring instruments, and also does not specify the types of devices.

4. The manuscript does not compare the obtained photovoltaic characteristics with the parameters of photodetectors known in the literature.

5. There is no line numbering in the manuscript.

The annotation must be rewritten taking into account the conversion of current into current density.

P1 L6-8. The text is unclear.

P2 L14-15. “ ..heterogeneous devices..” – what are these devices?

P2 L32-41. The description is unclear and does not correspond to Fig.1. How was the clamping force of one material to another observed?

P3. “..heterogenic structure..” – the expression is incomprehensible.

P3. “..atomic mode..” – the expression is incorrect.

P3. “…Class I semiconductor heterojunction.” – What is this semiconductor heterojunction?

Fig.2 – Fig.2,a –This figure is unclear and does not reflect the structure of the resulting device. Fig.2b – The figure contains a gap between Ta2NiSe5 and WSe2. Is this gap really there? Fig. 2c,d – Energy band diagram are drawn extremely poorly.

What is the work of the electron output from Ta2NiSe5 and WSe2?

Fig.3a, 4a, 5a shows the type of device. But this view in the drawings is different. Somewhere there are electrode designations, somewhere there are not. This drawing needs to be redone.

In Fig. 3a, 4a,4b, 5b, 5c –the current axis (I) must be passed through "0".

P4. It is necessary to specify the types of high-speed oscilloscopes.

P4 L6-19. It is necessary to describe the measurement procedure more clearly. Which electrodes were the voltage applied to? In what place was the laser shining? Why the image of the spot in Fig. 3c is shown outside the contact pads. How then did the electric field separate the charge carriers? Where does the electric field come from here if there is no heterojunction in the place where the laser shines?

Fig. 3C, 4c, 5d, 5e is a short-circuit current?

Fig.3d - the signature of the drawing and the drawing itself are unclear.

Fig.4 and the description to it, in my opinion, are useless, since they show a slight reaction to radiation. The authors did not specify the measurement error. The difference in the current when applying radiation with a power of 231 mW and 1405 mW is only 3%. This is within the measurement error. I think that this drawing and its description can be removed from the article. The reader only takes time to understand the result.  This graph shows only the disadvantage of the device being formed.

In Fig. 4a the electrical voltage is in the range from -0.1 to 0.1 V. And the text indicates 1 V. Where is the error?

P6 L7. Where the authors saw "...semiconductor junction...". They're talking about heterojunction.

P6 L7-9. The text is unclear. If the authors want to show that they have obtained photovoltaic characteristics, then it is necessary to specify the appropriate parameters - short-circuit current and open-circuit voltage. It is necessary to build an appropriate graph showing these parameters.

P6 L28-31 and fig.5i. The response time at 90% is determined when the signal increases from 0% to 90% of its maximum value, and not from 10% to 90%. Same for the recovery time. The link [41] is incorrect.

P7. The description of the experiment and Figure 6 are insufficiently described. Therefore, it is unclear why this experiment was carried out.

Round 2

Reviewer 1 Report

The authors have modified the manuscript according to the suggestions and hence can be accepted for final publication.

Reviewer 2 Report

Reviewer’s comments:

1.    In Abstract section, the critical research results and data such as response rate, responsivity, sensitivity, detectivity, and dark current should be exhibited.

2.    In Table 1, the typing format should be revised including the superscript and subscript in number. The cited literatures should be also typed in [50], but not "50" etc.

Reviewer 3 Report

The authors responded to most of the comments.

However, they did not respond to some of my comments:

1. The use of electric current density instead of electric current values makes it possible to make the measured parameters universal, improve the manuscript and make it more quotable. To do this, you just need to calculate the contact area of the heterojunction. The authors have geometric dimensions of the contact, but they don't want to do it. Why? A link to other good articles is not enough.

2. As a rule, on the IV dependencies, the axes pass through the coordinate (0,0). That is (I=0 A; U=0 V). Then the short-circuit current and the no-load voltage are clearly visible on the curves. Perhaps you need to show an increased bid. Then the drawings will look beter.
